# Antibiotic Resistance Mediated by *Escherichia coli* in Kuwait Marine Environment as Revealed through Genomic Analysis

**DOI:** 10.3390/antibiotics12091366

**Published:** 2023-08-25

**Authors:** Hanan A. Al-Sarawi, Nazima Habibi, Saif Uddin, Awadhesh N. Jha, Mohammed A. Al-Sarawi, Brett P. Lyons

**Affiliations:** 1Environment Public Authority, Fourth Ring Road, Shuwaikh Industrial 70050, Kuwait; 2Environment and Life Science Research Centre, Kuwait Institute for Scientific Research, Safat 13109, Kuwait; sdin@kisr.edu.kw; 3School of Biological Sciences, Plymouth University, Drake Circus, Plymouth PL4 8AA, UK; a.jha@plymouth.ac.uk; 4Department of Earth & Environmental Sciences, Kuwait University, Faculty of Science, P.O. Box 5969, Safat 13060, Kuwait; sarawi500@gmail.com; 5Research & Monitoring Coordination Nature Conservation Department, Neom 49625, Saudi Arabia; brettlyons1@hotmail.com

**Keywords:** class 1 integrons, antibiotic resistance genes, horizontal gene transfer, gene cassette, plasmids, marine environment

## Abstract

Antibiotic-resistance gene elements (ARGEs) such as antibiotic-resistance genes (ARGs), integrons, and plasmids are key to the spread of antimicrobial resistance (AMR) in marine environments. Kuwait’s marine area is vulnerable to sewage contaminants introduced by numerous storm outlets and indiscriminate waste disposal near recreational beaches. Therefore, it has become a significant public health issue and warrants immediate investigation. Coliforms, especially Gram-negative *Escherichia coli*, have been regarded as significant indicators of recent fecal pollution and carriers of ARGEs. In this study, we applied a genome-based approach to identify ARGs’ prevalence in *E. coli* isolated from mollusks and coastal water samples collected in a previous study. In addition, we investigated the plasmids and *intl1* (class 1 integron) genes coupled with the ARGs, mediating their spread within the Kuwait marine area. Whole-genome sequencing (WGS) identified genes resistant to the drug classes of beta-lactams (*bla_CMY-150_*, *bla_CMY-42_*, *bla_CTX-M-15_*, *bla_DHA-1_*, *bla_MIR-1_*, *bla_OKP-B-15_*, *bla_OXA-1_*, *bla_OXA-48_*, *bla_TEM-1B_*, *bla_TEM-35_*), trimethoprim (*dfrA14*, *dfrA15*, *dfrA16*, *dfrA1*, *dfrA5*, *dfrA7*), fluroquinolone (*oqxA*, *oqxB*, *qnrB38*, *qnrB4*, *qnrS1*), aminoglycoside (*aadA2*, *ant(3’’)-Ia*, *aph(3’’)-Ib*, *aph(3’)-Ia*, *aph(6)-Id*), fosfomycin (*fosA7*, *fosA_6*, *fosA*, *fosB1*), sulfonamide (*sul1*, *sul2*, *sul3*), tetracycline (*tet-A*, *tet-B*), and macrolide (*mph-A*). The MFS-type drug efflux gene *mdf-A* is also quite common in *E. coli* isolates (80%). The plasmid *ColRNAI* was also found to be prevalent in *E. coli.* The integron gene *intI1* and gene cassettes (GC) were reported to be in 36% and 33%, respectively, of total *E. coli* isolates. A positive and significant (*p* < 0.001) correlation was observed between phenotypic AMR-*intl1* (r = 0.311) and phenotypic AMR-GC (r = 0.188). These findings are useful for the surveillance of horizontal gene transfer of AMR in the marine environments of Kuwait.

## 1. Introduction

The excessive use of antibiotics to treat infectious diseases has legated the world to the public health hazard of antibiotic resistance [1,2,3]. Aquatic environs have been identified as reservoirs of antibiotic resistancegenes (ARGs); however, little is known about their distribution, spread and migration [4,5]. Marine environments are also sinks for pharmaceuticals, disinfectants, heavy metals, organic compounds, microplastics, and atmospheric dust [6,7,8,9,10,11]. Among these, pollutants, pharmaceuticals, and heavy metals have been evidenced to impose selective pressure on inherent bacterial communities that often develop resistance genes against them [9,11,12,13,14,15,16,17,18]. In addition, genetic elements, for example, class 1 integrons, package ARGs into gene cassettes (GCs) and mediate their transfer to non-resistant bacterial communities via horizontal gene transfer (HGT) [19,20,21,22]. Although more consideration has been given to animal and human health with respect to antimicrobial resistance (AMR) monitoring, the environment often remains ignored. The World Health Organization propelled the One Health Concept, which calls for the evaluation of overall environmental health [23]. Immediate action is also warranted to monitor the threats of AMR in the Gulf Cooperation Council [24,25].

Emergency waste and unauthorized sewage discharge in Kuwait have introduced several antibiotic-resistant bacteria (ARBs) into Kuwait’s marine environments [18,26]. Among these, *Escherichia coli* could become an infectious animal and human bacterium showing resistance to almost all clinically used antibiotics. A wide profile of AMR *E. coli* has been previously demonstrated through conventional phenotypic antibiotic susceptibility testing (AST) in Kuwait’s marine environment [27,28,29]. However, data on the genes conferring resistance are lacking in these aquatic settings. In addition, class 1 integrons, GCs, and plasmids involved in their spread are unknown. Currently, microbiological methods of AMR surveillance are considered laborious, time-intensive, and provide limited phenotypic information. With the advent of molecular methods, a rapid and detailed genotypic assessment of all these components can be achieved [30,31,32,33,34]. These methods are now being embraced as a novel approach to map the whole genomes of bacterial isolates, ARGs, plasmids, integrons, and GCs simultaneously to complement the AMR phenotypic assays [32,35,36,37]. The goal of this research was to examine the genomic profiles of *E. coli* through whole-genome sequencing and to study the prevalence of ARGs, plasmids, integrons (*intl1*), and GCs. 

## 2. Results

For the present investigation, selected *E. coli* isolates (n = 23) were subjected to whole-genome sequencing. All sequences ranged between 4,577,350 and 5,103,695 bases, with a Phred quality score above 20. 

### 2.1. MLST and Phylogenetic Analysis

The open reading frames corresponding to seven sequence tags (STs), namely, *adk-fumC-gyrB-icd-mdh-purA-recA*, were identified in the assembled genomes of 23 *E. coli* isolates. Alignment of sequences against these STs established the *Enterobacteriaceae* origin of the strains. Further treatment with the mafft software and UGENE software confirmed 20 isolates as *E. coli*, and three of these closely matched with *Enterobacter cloacae* (SE111, SE181, and SE158). Molecular data analyses helped to correctly identify the species of *E. cloacae*; hence, we found the molecular tests to be technically superior for discriminating between closely related *E. coli* and *E. cloaca* strains. Further analysis was performed on the 20 *E. coli* isolates. The phylogenetic relationships of the selected *E. coli* were distributed into two clusters (Figure 1). A close relationship was observed between the isolates from spatially distant locations, different seasons, and varied matrices (marine waters or mollusks). For example, SE25 and KHE11 isolated from Al-Salam and Khiran were closely related. Strains SE19 and SC118 were isolated from different matrices, i.e., marine water during winter and mollusks collected during summer; however, they were positioned on the same branch of the phylogenetic tree for their similarity. Similarly, SC59 and 756E0 were also analogous. It was noticed that the three isolates of *E. cloacae*, namely, SE111, SE181, and SE158, exist separately as a group. 

### 2.2. Antibiotic-Resistance Gene Elements

We submitted the assembled *E. coli* sequences to the ResFinder database and found matches against 33 ARGs (Figure 2; Appendix A). The percentage identity of the sequences ranged between 80 and 100%. The highest frequency (80%) was that of the *mdf (A)* gene found in at least 16 isolates. This was followed by *bla_Tem-1B_*, present in 45% of *E. coli* strains. 

The highest number of genes (10/33) originated from beta-lactam family (*bla_CMY-150_*, *bla_CMY-42_*, *bla_CTX-M-15_*, *bla_DHA-1_*, *bla_MIR-1_*, *bla_OKP-B-15_*, *bla_OXA-1_*, *bla_OXA-48_*, *bla_TEM-1B_*, *bla_TEM-35_*). This was followed by trimethoprim (6/33),-(*dfrA14*, *dfrA15*, *dfrA16*, *dfrA1*, *dfrA5*, *dfrA7*), fluoroquinolone (5/33) (*oqxA*, *oqxB*, *QnrB38*, *QnrB4*, *QnrS1*),aminoglycoside (5/33) (*aadA2*, *ANT (3’’)-Ia*, *APH (3’’)-Ib*, *APH (3’)-Ia*, *APH(6)-Id*), fosfomycin (4/33) (*FosA7*, *FosA6*, *FosA*, *FosB1*), sulfonamide (2/33) (*sul1*, *sul2*), and tetracycline (2/33) resistance genes (*tetA*, *tetB*). Single genes belonged to the drug class macrolide (*mphA*) and MFS-type drug efflux (*mdfA*). 

We compared the ARG summaries of these 20 strains with their AMR phenotypes (tested against a panel of 23 antibiotics). According to the genotypic assay, all these strains possessed ARGs; therefore, they were considered as potentially resistant to at least one of the drug classes. Phenotypically, the strains SC59, HE33, 7SE60, SE124, and CW140 were sensitive to all the antibiotics (Figure 3). 

Resistance to beta-lactams was confirmed both genotypically and phenotypically in 40% of the isolates, namely, HE40 (*bla_TEM-1B_*), SC63 (*bla_TEM-1B_*), SC70 (*bla_OXA-1_*, *bla_OXA-48_*, *bla_TEM-35_*), SC118 (*bla_CMY-42_*, *bla_TEM-1B_*), SE25 (*bla_CTX-M-15_*), SE19 (*bla_TEM-1B_*), KHE11 (*bla_CMY-150_*, *bla_TEM-1B_*), and CW138 (*bla_TEM-1B_*). Strain SE138 bearing beta-lactamase genes (*bla_DHA-1_*, *bla_MIR-1_*, *bla_TEM-1B_*) depicted an intermediate phenotype against this drug class. Strain 8KHE1 was phenotypically resistant to beta-lactams + beta-lactamase inhibitors and cephalosporins. None of the genes were resistant to the above drug classes; rather, the *mdfA* (MFS-type drug efflux) resistant against tetracycline, disinfectants, and antiseptics was found in the genotype. Similar was the case with strains SC117 and SC89. Within the beta-lactamase, genes resistant to CMY-beta lactamase (cephamycin), CTX-M-beta lactamase (penam, cephalosporin), DHA-beta lactamase (cephalosporin, cephamycin), MIR-beta lactamase (monobactam, cephalosporin), OKP-beta lactamase (penam, cephalosporin), OXA-beta lactamase (penam, cephalosporin, carbapenem), and TEM-beta lactamase (penam, monobactam, penem, cephalosporin) were observed in strains SC118; SE25, SE138; SE138; SE124; SC70; and SC59, HE40, SC63, SC70, SC118, SE19, KHE11, SE138, CW138, and CW140, respectively. Cephalosporin was one of the drugs in the antibiotic panel. As evident from the above statements, this first-generation beta-lactamase confers resistance against all the sub-categories of beta-lactams. TEM-beta lactamase was one of the most common genes detected in the tested isolates (45%). Among the carbapenems, the SE19 isolate expressed a meropenem (carbapenem)-resistant phenotype and also possessed bla_TEM_ (TEM-beta lactamase) gene. All the remaining strains were susceptible to meropenem. In addition to this, susceptibility to imipenem (carbapenem) was observed in all the strains (Figure 3). None of the genotypes were positive for the *bla_IMP_* group of genes. Strain CW141, phenotypically resistant to piperacillin (beta-lactams), was devoid of relevant genes.

Matching genotypes and phenotypes (20%) for aminoglycoside resistance were recorded in HE40 (*aph(6)-Id*), HE73 (*aph(3’’)-Ib*, *aph(6)-Id*), KHE11 (*ant(3’’)-Ia*), and SE138 (*aph(3’’)-Ib*, *aph(6)-Id*), whereas genes without any phenotypic expression for this drug class were observed in SC59 (*aph(3’’)-Ib*, *aph(3’’)-Ib*), SC70 (*ant(3’’)-Ia*), CW138 (*aph(3’’)-Ib*, *aph(6)-Id*), and CW140 (*aph(3’’)-Ib*, *aph(6)-Id*). Strains 8KHE1, SC117, and SC89 depicted an aminoglycoside-resistant profile but were devoid of related genes. Contrastingly, its genotype possessed *mdfA* (MFS-type drug efflux) resistance against tetracycline, disinfectants, and antiseptics. 

In the case of fluroquinolones, phenotypic resistance was expressed in 45% of isolates (HE40, SC63, SC117, SC118, HE733, SE19, KHE11, SE138, and CW138). Among these, only 15% (3/20) of the isolates, i.e., HE40, KHE11, and SE138, possessed resistance genes (*qnrS1*). Phenotypic sensitivity against quinolone (used interchangeably with fluroquinolone) was demonstrated by all the strains except KHE11. ARGs against this drug class were located in strains SC59 (*qnrS1*), HE40 (*qnrS1*), SC70 (*qnrS1*), SE25 (*qnrS1*), and SE138 (*oqxA*, *oqxB*, *qnrB4*). 

Trimethoprim- or sulfonamide-resistant phenotypes were observed in KHE11, CW138, CW121, and CW141, of which only the first two strains (10%) possessed ARGs (KHE11—*dfrA15*, *sul1;* CW138—*dfrA7*, *sul2*). Four strains with intermediate phenotypes were positive for corresponding ARGs (HE40—*dfrA14*, *sul2*; SC63—*sul2*; HE73—*sul2*; and SE138—*dfrA15*, *sul1*, *sul2*).

Moreover, SC59, HE40, SC63, SC70, HE73, SE25, SE19, KHE11, SE138, CW138, and CW140 all possessed tetracycline-resistant genotypes (ARG—*tetA*, *tetB*), which was not captured by the AST as the chosen panel lacked tetracycline. Similarly, macrolide-resistant genes (*mphA*) were recorded in SC70 and fosfomycin-resistant genes (*fosB1*) in CW141 that were not tested phenotypically.

The alignment of sequences against the plasmid finder database revealed 13 isolates as hosts to 22 plasmids (Appendix A). The most prevalent was the *ColRNAI* plasmid found in 11 isolates (42.3%) (Figure 4). Sequences were then concomitantly mined for integrons, recombination sites, and promoters. The genetic elements identified were *attC* (gene cassette recombination sites), *intI* (intersection tyr-integrase), *Pc_1* (gene cassette promoter class), *Pint_1* (integron promoter), and *attI_1* (integron recombination site). The elements *attC* and *intI* were reported in 20% (4/20) of the strains, whereas *Pc_1* and *attI_1* were detected in 10% (2/20) of the isolates. *Pint_1* was found in 15% (3/20) of the tested samples. Isolates CW138, CW121, and CW141 were positive for integron with the integron integrase (*intI*) and *attC* site nearby, whereas HE70 was positive for the latter only, lacking recombination sites (Table 1). These elements aid the mobilization of ARGs in other strains. The *intl1* gene encodes the integrase enzyme, the promoter Pc ensures its expression, and the site-specific recombination takes place between the *attl* and *attC* sites. The GC is an open reading frame (ORF) without a promoter but with a recombination site. These GCs can integrate novel genes and support the bacterial strains to adapt. On the other hand, strain SC70 possessed only a cluster of *attC* sites without any integrase in its vicinity (Appendix A). Classification of integrons and gene cassettes revealed the presence of class 1 integron (*intl1*) and GC1. All the genetic elements within strain CW141 are shown in Figure 5. Intriguingly, this strain possessed the fosfomycin-resistant gene (*fosB1*) and expressed phenotypic AMR against piperacillin (an extended spectrum beta-lactam) and trimethoprim, as well as harbored the incX1 plasmid.

### 2.3. PCR-Based Identification of Intl1 and GCs 

Among all the isolated strains (n = 598 from both mollusks and seawater), 216 were positive for the *intl1* (36%) gene, exhibiting a band of 400 bp in size on the agarose gel. These were associated with 198 GCs (36%) (Figure 6A). We looked at the AMR phenotypes of *intl1-*positive isolates (n = 216); interestingly, 95% of them were phenotypically resistant to at least one antibiotic among the panel (aminoglycoside, beta-lactams, aminoglycoside + beta-lactam inhibitors, beta-lactams/beta-lactamase inhibitors, cephalosporins, fluroquinolones, quinolones, imipenem, and dihydrofolate-reductase/trimethoprim) and the remaining 5% were susceptible (Figure 6C) to all the tested antibiotics. Considering the total AMR phenotypes of *E. coli* (n = 420), approximately 49% were negative for *intl1* (Figure 6B). Spearman’s correlation model established a positive correlation between phenotypic AMR-*intl1-*GC. Overall, the coefficients were statistically significant (*intl1* ß–0.230, *p* < 0.001; GC ß–0.121, *p* < 0.001). There was 7.5% variance between *intl1-*GC resistance (F—24.28, *p* < 0.001). Positive correlations were observed between phenotypic AMR-*intl1* (0.310; *p* < 0.001) and phenotypic AMR-GC (0.190; *p* < 0.001). The *intl1* and GC were also significantly correlated (0.120; *p* < 0.003).

## 3. Discussion

WGS has become the most preferred method to identify ARGs and associated genetic elements [38]. Recently, it identified ARGs in both highly resistant and susceptible isolates in the order of Enterobacterales [38]. This indicates that a population of resistant *E.coli* is floating in the coastal waters of Kuwait [36,39]. In addition to resistance profiling, WGS was also used presently for the identification of *Enterobacteriaceae* species. Interestingly, three of the isolates (13.0%) were identified as *Enterobacter cloacae*. A study from the Norwegian coast demonstrated that MALDI-TOF-MS (Matrix-assisted laser desorption ionization–Time of flight mass spectrometry) identified 90% of isolated strains as *E. coli*, whereas the remaining 10% were *Klebsiella*, *Citrobacter*, and *Enterobacter* [40]. More robust and precise identification approaches such as molecular methods are preferred over conventional microbiological assays, especially in mollusks and other filter feeders in the marine environment that are prone to pathogenic exposure, in addition to *E. coli*, for accurate discrimination between closely related species [41]. Further, the MLST-based phylogeny established a close relationship between the 20 isolates. This indicates a common origin of *E. coli*, most probably one of the point sources of emergency outfalls. The genetic similarity between the *E. coli* isolates of seawater and mollusks is most probably due to the filter feeding mechanism of the latter [42]. A comprehensive analysis might assist in contact tracing of these AMR strains. 

Intriguingly, the *E. coli* isolates (n = 20) possessed genes resistant against seven drug classes. This is most likely due to the selective pressure imposed by antibiotics such as Azithromycin, Cefalexin, Ciprofloxacin, Clarithromycin, Dimetridazole, Erythromycin, Metronidazole, Metronidazole-OH, Ofloxacin, Ronidazole, Sulfamethoxazole, Tetracycline, and Trimethoprim found in seawater samples [8] and the effluent streams of waste-water treatment plants in Kuwait [43]. Moreover, the presence of conjugative plasmids and integrons in these strains is indicative of ARG transmission through horizontal gene transfer [19,20,22,44,45]. Nevertheless, this study only targeted the fecal coliforms, and the likelihood of other microbial communities with diverse ARG profiles is expected. The advanced method of high-throughput shotgun metagenomic sequencing would be a more valuable tool in the assessment of resistomes at contaminated and non-contaminated sites [26,38].

The presence of beta-lactam genes and their phenotypic expression in strains such as HE40, SC63, SC70, SC118, SE25, SE19, KHE11, SE138, and CW138 draws our immediate attention. Moreover, plasmid vectors were also recorded in HE40, SC70, SE25, SE19, SE138, and CW138. In addition, the strains HE40 and CW138 also possessed integrons and gene cassettes, indicating the possibility of horizontal gene transfer of beta-lactamase genes in the region. Extended spectrum beta-lactams are the last-choice drugs to treat hardcore bacterial infections [46,47]. Worldwide circulation of beta-lactamases have been reported recently with more than 40% in Asian countries [48]. A baseline study also documented the presence of beta-lactams in aerosols collected from a public building situated near the coast of Kuwait [49]. Moreover, trimethoprim resistance was confirmed in strains KHE11 and CW138. The dominance of the MFS-type drug efflux gene (tetracycline, disinfectants, antiseptics) was also noticed in 75% of sequenced isolates. Tetracycline-resistant genes (*tetA/tetB)* were found in 55% of isolates. The AMR phenotypes for these antibiotics were missing and need to be taken into consideration while performing future phenotypic assays.

We also report that 35.99% of AMR *E.coli* held the *intl1* gene, in parallel with *Enterobacteriaceae* found in Korea (41.4%) [50]. Contrastingly, only 12.1% of *Enterobacteriaceae* strains were positive for class 1 integrons in an effluent handling system in Poland [51]. *Intl1* positive strains were maximum in locations proximal to the coastal emergency waste outfalls. Numerous storm outlets discharge emergency waste into the country’s marina, leading to deteriorated water quality [24,27,52,53]. These findings are in congruence with our earlier reports on the presence of AMR microbes, ARGs, integrons, plasmids, and insertion sequences near the outfalls [18,26]. We attribute this to the antibiotics, pharmaceuticals, and other contaminants introduced into the seawater [8,43]. A constant selection pressure is thought to elevate the horizontal gene transfer frequency within these aquatic milieus [25,33,36,45,54,55]. Regression analysis predicted a relationship (7.5% variance) between *intI1*-GC resistance. This suggests that the class 1 integron could potentially play a role in the transmission of AMR in *Enterobacteriaceae* [45]; however, other contaminants, such as metals, cannot be ignored [5,56,57,58].

In the present investigation, WGS identified ARGs in both resistant and susceptible *E. coli* phenotypes. The presence of *intl1* and GCs indicates the HGT of AMR in the region. WGS is recommended as an accurate and precise monitoring tool for individual AMR strains perpetrating in marine ecosystems.

## 4. Materials and Methods

### 4.1. Sampling Locations, Collection and Isolation of Strains

A detailed overview of the sampling task is provided elsewhere [28] as they were previously collected and tested for AMR through standard microbiological methods. However, for ease of readership, a succinct description is provided in Section 4.1.1, Section 4.1.2 and Section 4.1.3. A detailed description of the genomic analysis performed in the current study has been described in Section 4.2, Section 4.3, Section 4.4 and Section 4.5.

#### 4.1.1. Marine Waters

Seawater was collected from Al-Ghazali (29°347784 N, 47°911974 E), Al-Salam (29°357207 N, 47°946784 E), Abu Al-Hasaniya (29°125652 N, 48°633155 E), and Al-Khairan (28°665070 N, 48°387640 E) in an earlier campaign [28]. Six replicates were sampled in 200 ml sterile bottles from each site in both summer (Jul–Aug 2015; 34.0 °C ± 2.0) and winter (Dec–Feb 2016; 17.2 °C ± 2.0) seasons (Appendix A). In situ measurements of pH (8.2 ± 0.2), salinity (42 ± 2.0), and temperature were made employing a hand-held, multi-probe water quality meter (Hanna instruments, Smithfield, RI, USA). Samples were transported on ice to the Environment Public Authority (EPA) Kuwait laboratories. Samples were processed at EPA laboratories following an established protocol [36]. Briefly, 10 mL of seawater was pipetted into ¼ strength Ringer’s solution and serially diluted (10 to 10^5^). Serial dilutions were passed through 0.45 µm Merck filters (Merck Life Sciences Limited, Dorset, UK). The filters were placed on Tryptone Bile X-Glucuronide chromogenic media (TBX) selective for *E. coli.* The Petri dishes were placed inverted at 30 °C for 4 h and 44 °C for 21 h. The plates were examined for the growth of blue–green colonies marking the presence of *E. coli*. A total of 351 strains were isolated from the collected water samples.

#### 4.1.2. Mollusk Samples

Concurrent to the seawater sampling, mollusk samples were gathered in the summer (Jul–Aug 2015; 34.0 °C ± 2.0) and winter seasons (Dec–Feb 2016; 17.2 °C ± 2.0). Shells (n = 55–60) were hand-picked, sorted in autoclaved bags (4 °C), and transported to the College of Science, Kuwait University laboratories for further processing [28]. The mollusc shells were removed to collect the flesh and fluid in a pre-autoclaved beaker (Borosil, Poole, UK). Approximately 20 g was macerated with 0.1% peptone water (Sigma, Plymouth, UK) and serially diluted [28]. The dilutions were passed through the 0.45 µm nitrocellulose membrane filters and placed on TBX-agar for *E. coli*, as mentioned previously. In total, 247 mollusc strains were isolated.

#### 4.1.3. Antimicrobial Susceptibility Testing

Antibiotic susceptibility assays for all these isolates were performed, and minimum inhibitory concentration (MIC) was recorded and presented in Al-Sarawi et al. [28]. Briefly, the isolates were tested for susceptibility and resistance against a panel of 23 antibiotics as per the guidelines of Clinical and Laboratory Standards Institutes [59]. The micro-dilution method was employed to determine the MIC by inoculating a loopful of isolate into 10 mL of Cation-Adjusted Muller Hinton (CAMBH) broth (incubated on shaker incubator for 24 h at 37 °C). Turbidity was adjusted with 0.5 M MacFarland solution and loaded to custom, dehydrated, 96-well Sensititre^TM^ plates (GN2F, Thermo Scientific, Paisley, UK), which were further incubated at 25 °C for 24 h. Strains were classified as resistant, intermediate, or susceptible based on the MIC breakpoints standardized by CEFAS [27].

### 4.2. Whole-Genome Sequencing (WGS) and Filtering of ARGEs

From the 598 isolates, selected strains (n = 23, Appendix A) with interesting AMR phenotypes, i.e., resistant to 2 or more antibiotics Al- Sarawi et al. [28] or susceptible to all, were subjected to WGS. These isolates were also chosen to represent varied spatiotemporal variability (three sites and two seasons). Whole-genome DNA was isolated from an over-nightly activated broth. Approximately 15 mL of Tryptic Soy Broth was seeded with a loopful of *E. coli* and incubated at 37 °C. The culture tube was centrifuged at 13,000× *g* to pellet the organisms and subjected to DNA purification as per the standard protocol of Promega Wizard ^®^Genomic DNA extraction Kit (Promega, Chilworth Southampton, Hampshire, UK) [60]. Fluorometric quantification of DNA was conducted on a Quantus fluorometer (Promega, Chilworth Southampton, Hampshire, UK). Sequencing libraries were prepared following the Nextera XT kit (Illumina, USA). The libraries were purified using AMPure^®^ XP (Beckman Coulter, Brea, CA, USA) magnetic beads. Libraries were normalized and pooled before loading on a MiSeq V3 sequencing cartridge. Sequencing was performed on an Illumina MiSeq platform available at the Centre for Environment Fisheries and Aquaculture Science (CEFAS), Weymouth laboratories through paired-end (2 × 300 bp) chemistry [36]. Post-sequencing, the Illumina adapters were removed by Trimmomatic v 0.36 [61] and a quality check was performed on the web version of FastQC v 0.11.5 [62]. Raw reads were thereafter aligned in Spades v 3.10.1 for prokaryotic assembly [63] and annotated via Prokka v1.11 [64]. The ABRicate tool [65] was used to screen the ‘assembled fasta’ for antibiotic-resistant genes (ARGs), employing the ResFinder v 4.0.0 [66] database (accessed on 2 April 2023). The sequences were also mined for plasmids through the plasmid finder database [67]. The integron finder database v 2.0.2 was used to annotate the complete integron with integron integrase, *attC* recombination sites, promoters, and nearby gene cassettes [68]. 

### 4.3. MLST (Muti-Locus Sequence Typing) and Phylogenetic Analysis

Seven MLST genes, namely, *adk-fumC-gyrB-icd-mdh-purA-recA* were retrieved from the Pfam archive. The respective ids of the gene sets were PF00406, PF05683, PF00204, PF00180, PF00056, PF00709, and PF00154, respectively [69]. The hmmscan v 3.1, [70] was then used to align the assembled *E. coli* genomes with the abovementioned MLSTs. The identified MLSTs were aligned in mafft v 7.305 using the default settings [71]. UGENE v 1.98 was employed to remove and concatenate the unaligned sequences from the 5′ and 3′ ends [72]. A maximum likelihood tree analysis was performed on the aligned MLSTs through FastTree v 2.1.8 [73]. A phylogenetic tree was plotted in FigTree v 1.4.2 [74]. 

### 4.4. Screening for Class 1 Integrons and Associated Gene Cassettes

A total of 598 *E. coli* pure cultures isolated from marine waters (n = 351) and marine organisms/mollusks (n = 247) were used for *intl1* gene screening [28,75] and associated gene cassettes (GC). DNA was isolated as per the protocol described elsewhere [76]. Primers int1.F/R (F-5′GGGTCAAGGATCTGGATTTCG 3′; R-5′ACATGCGTGTAAATCATCGTCG3′) and HS286 (F-5′ GGGATCCTCSGCTKGARCGAMTTGTTAGVC) HS287 (R-5′GGGATCCGCSGCTKANCTCVRRCGTTAGSC3′) were used to amplify the *intl1* and GCs, respectively [33,77]. A PCR mix (25 μL) was prepared by combining 2 μL (4 µM) each of forward and reverse primers, 6.5 μL of PCR grade water, 12.5 µL of buffer (MangoMix^TM^, Bioline, London, UK), and 2 μL of template DNA. The reaction was run for initial denaturation (94 °C for 60 s) followed by 35 repetitions (denaturation—94 °C, 60 s; annealing—58 °C for 30 s; extension—72 °C for 150 s) and final extension at 72 °C on a Gene Amp PCR system 9700 (Applied Biosystem, Cheshire, UK). PCR products were passed through a 1.5% agarose gel (Fisher, NJ, USA) and visualized on a GelDoc XR gel imaging system (Bio-Rad, Hertfordshire, UK). 

### 4.5. Statistical Analysis

Statistical analysis was performed in SPSS 23.0 (IBM, SPSS, Hampshire, UK). The normality of data was tested through the Shapiro–Wilk test. A *t*-test was run to check for the no-normality violation and homogeneity assumptions. Data were considered significant at a confidence interval of 99.95% (*p* < 0.05). Regression tests applying rectilinear models were employed to establish the correlation between *intl1-*GC resistance. Homogeneity of variance, data normality, and dimensions of linearity were also explored.

## Figures and Tables

**Figure 1 antibiotics-12-01366-f001:**
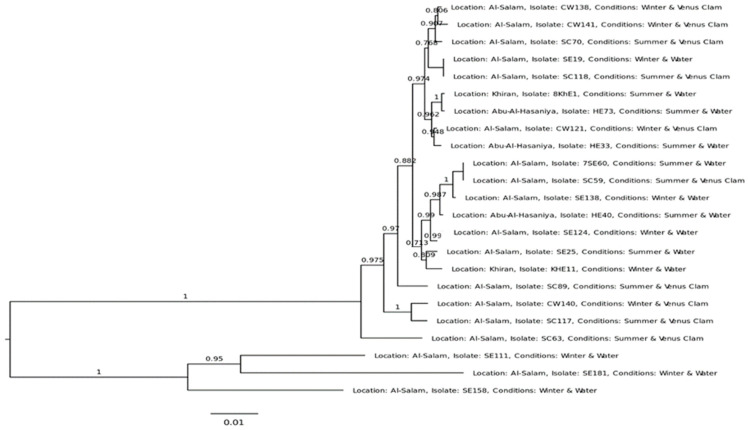
Phylogeny of the strains isolated from Kuwait. A maximum-likelihood tree was plotted on the distances between the MLST sequences of the selected strains.

**Figure 2 antibiotics-12-01366-f002:**
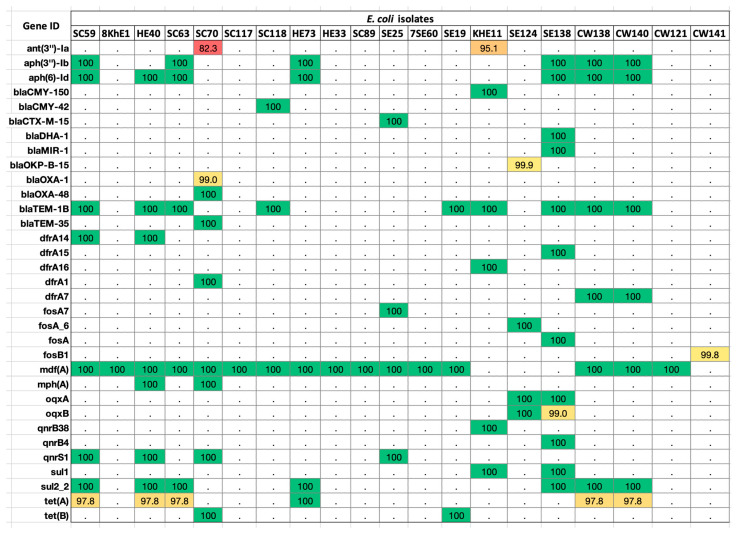
ARGs distributed in the *E. coli* genomes. Colored boxes are positive for the respective ARG listed on the left-hand side panel. The values in each box show the percentage identity in the ResFinder database v4.0.0 (red > 80%; yellow > 90%; green > 100%; white < 0.0%).

**Figure 3 antibiotics-12-01366-f003:**
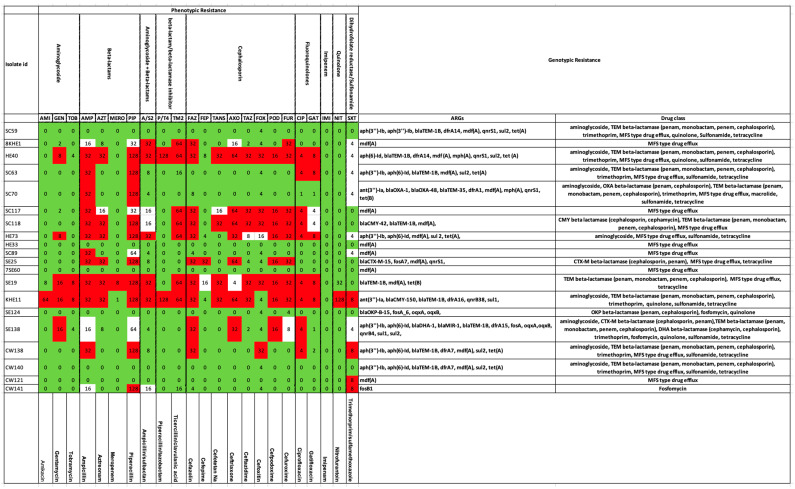
Genotypic versus phenotypic resistance in selected isolates of *E. coli.* Phenotypes are presented on the left-hand side panel. The color codes represent Red—Resistant, Green—Susceptible, and White—Intermediate. Corresponding genotypes are shown on the right-hand side.

**Figure 4 antibiotics-12-01366-f004:**
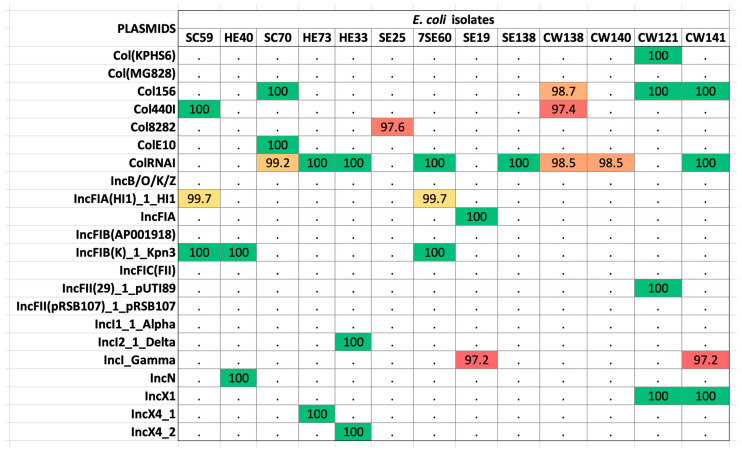
The plasmids found in selected isolates of *E. coli* revealed by whole-genome sequencing. Colored boxes mark the presence of the plasmids and their percentage identity in the database (red > 80%; yellow > 90%; green > 100%; white < 0.0%). Plasmids were fetched from the PlasmidFinder database.

**Figure 5 antibiotics-12-01366-f005:**
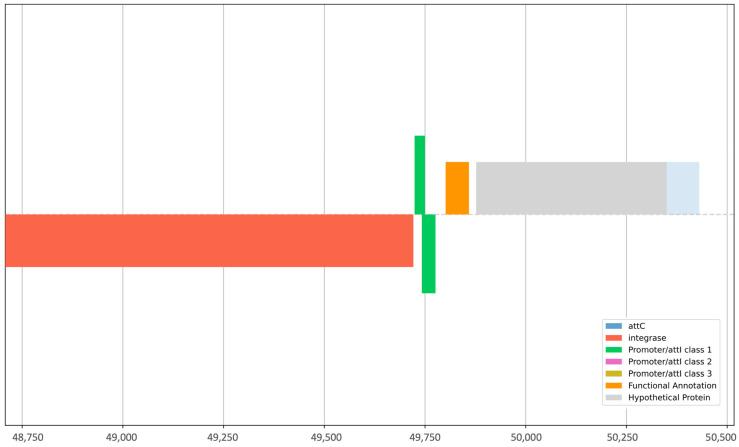
Graphical representation of integrons, recombination sites, and promoters present in *E. coli* strain CW141.

**Figure 6 antibiotics-12-01366-f006:**
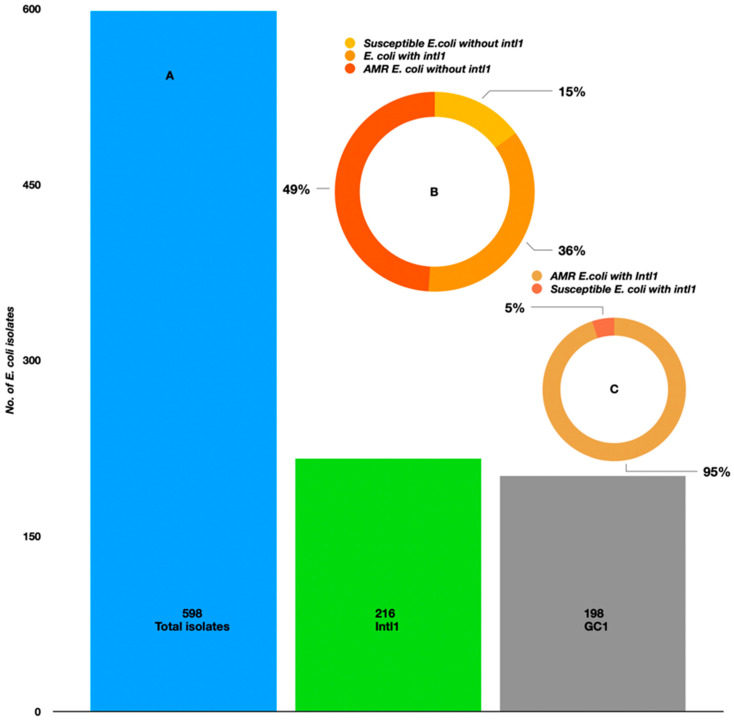
(**A**) Prevalence of class 1 integrons and gene cassettes in *E. coli* isolated from the Kuwait marine waters and molluscans. (**B**) The occurrence of class 1 integrons in *E. coli* isolated during summer and winter seasons. (**C**) AMR-susceptible *E. coli* with *intl1*.

**Table 1 antibiotics-12-01366-t001:** Integrons in *E. coli* isolates.

Sample	Elements Identified by Integron Finder					Integron Class	
	*attC*	intI	*Pc_1*	*Pint_1*	*attI_1*	*intl1*	GC
HE40	-	Yes	Yes	-	Yes	-	-
CW138	Yes	Yes	-	Yes	-	Yes	Yes
CW121	Yes	Yes	-	Yes	-	Yes	Yes
CW141	Yes	Yes	Yes	Yes	Yes	Yes	-
SC70	Yes	-	-	-	-	-	-

*attC*—GC recombination site; intI—intersection tyr-integrase; *Pc-1*—gene cassette promoter int1; *Pint_1*—promoter intl1; *attI_1*—integron recombination site; *intl1*—integrase gene; GC—gene cassette.

## Data Availability

Raw Sequences are hosted on the web portal of the National Centre for Biotechnology Information (NCBI) under accession no. PRJNA955288 (SRR24162651 to SRR24162673) https://dataview.ncbi.nlm.nih.gov/object/PRJNA955288?reviewer=gfeu7sag3l3hgq6o8k49nk974 (accessed on 15 June 2023).

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
