# Peer review of "Antibiotic Resistance Mediated by Escherichia coli in Kuwait Marine Environment as Revealed through Genomic Analysis"

_antibiotics, 2023, doi:10.3390/antibiotics12091366_

Round 1

Reviewer 1 Report

Comments#

In the article entitled “Antibiotic Resistance Mediated by Gram Negative Escherichia coli in Kuwait Marine Environment as Revealed by Genomic Analysis” Sarawi et.al investigated the distribution of antibiotic resistance gene elements (ARGEs) among the E. coli strains isolated from the Kuwait marine environment- particularly the contamination prone areas. They performed a comprehensive sampling for the isolation of E. coli strains from the water as well as the mollusks and selected a total of 23 isolates resistant to antibiotics and were subjected to whole-genome sequencing. ResFinder was used to identify the ARGEs in the genomes and detected the presence of genes immune to beta-lactams and many others.

The study is well-designed, and the manuscript is well-written.

In the title, ‘Gram Negative’ requires a hyphen when it is used as a modifier.  

Make sure to italicize the scientific names of the organisms throughout the manuscript.

Lines 77-80 – It is true that the isolates from marine water and mollusks are clustered together.  Considering the filter-feeding mechanism of mollusks, it is not uncommon to observe this trend. Please consider this and add subsequent information in the discussion part.

Moreover, mollusks also harbor symbiotic/opportunistic/pathogenic bacteria, which could also lead to the misinterpretation of the data. A proper discussion assessing this will strengthen the current finding presented in the article.

Minor checks are required.

Author Response

In the article entitled “Antibiotic Resistance Mediated by Gram Negative Escherichia coli in Kuwait Marine Environment as Revealed by Genomic Analysis” Sarawi et.al investigated the distribution of antibiotic resistance gene elements (ARGEs) among the E. coli strains isolated from the Kuwait marine environment- particularly the contamination prone areas. They performed a comprehensive sampling for the isolation of E. coli strains from the water as well as the mollusks and selected a total of 23 isolates resistant to antibiotics and were subjected to whole-genome sequencing. ResFinder was used to identify the ARGEs in the genomes and detected the presence of genes immune to beta-lactams and many others.The study is well-designed, and the manuscript is well-written. 

 Response: We highly appreciate the reviewers support to this manuscript. We have attempted to further improve the revised version.

In the title, ‘Gram Negative’ requires a hyphen when it is used as a modifier.  

Response: We have removed the Gram Negative from the title.

Make sure to italicize the scientific names of the organisms throughout the manuscript.

Response: Noted with thanks. The scientific names are italicized in the revised manuscript.

Lines 77-80 – It is true that the isolates from marine water and mollusks are clustered together.  Considering the filter-feeding mechanism of mollusks, it is not uncommon to observe this trend. Please consider this and add subsequent information in the discussion part.

Moreover, mollusks also harbor symbiotic/opportunistic/pathogenic bacteria, which could also lead to the misinterpretation of the data. A proper discussion assessing this will strengthen the current finding presented in the article.

Response: We agree with the reviewer’s observation the filter feeding behavior is quite likely the reason for similarity between seawater and mollusks. We have now provided a brief description of both the reviewer’s suggestions in the revised discussion

Reviewer 2 Report

The manuscript describes genomic antimicrobial resistance profiles of E. coli of marine origin (water and molluscs), focusing on ARG, intI1, and plasmid detection. Despite the importance and soundness of the argument, the manuscript is confusing and hard to understand.

In the manuscript, authors describe E. coli as "non fecal coliforms", however they underline how sewage contaminants could affect marine environment. E. coli are usually considered faecal bacteria. I suggest to remove "non fecal coliforms" or to explain what authors refer to.

Kindly check the capital letter, cursive, commas, etc

Title

I suggest to remove "Gram Negative", because it seems redundant as all E. coli are Gram negative bacteria.

Abstract

line 22: substitute "prevalent" with "prevalence"

            "from the coast of Kuwait": clearly define the kind of samples                           analysed in the study (molluscs, water ecc)

line 23: substitute "its" with "their"

line 24: relocate "(WGS)" after "whole genome sequencing" and before "identified"

line 31: GCs, insert the extended name and the acronym in brackets.

line 32: acronym AMR, already defined in line 16.

line 32-33: I can't understand what authors mean by "significant correlation between AMR and GCs". Do you mean between phenotypic AMR and ARG presence?

Introduction

line 44-45: "These" is referred to pharmaceuticals...microplastics and atmospheric dust. However, only pharmaceuticals and heavy metals could impose selective pressure and lead to ARGs spread. Please, rephrase.

line 58: define "conventianal". Do you mean phenotypic antimicrobial susceptibility testing?

line 59-61: Do you refer to the water environment? Please, specify it. 

line 61: "conventional", as above.

line 63-63: I'm not sure you can say that molecular methods are superior to phenotypic ones. Phenotypic and molecular methods gave different and complementary information, all important for a better comprehension of the AMR phenomenon. Please, rephrase.

line 64: please define AST

Materials and methods

paragraph 4.1.1 Marine waters: How many samples of water do you have analysed? In which period? Please indicate how many E. coli do you select from each sample (mollusc, water), for example "From each sample, at least 2 presumptive E. coli were selected...."

line 210: What do you mean by "n=55-60 samples"? Do you refer to samples collected in summer (55) and winter (60)? Please indicate how many E. coli do you select from each sample.

line 218: Are these strains collected in another study or in the present one? In the first case, why do you choose to include these strains in the study? Which are their characteristics? Which antimicrobials were they resistant to? Were they isolated from water or molluscs? Please describe their features in a separate paragraph (4.1.3). In the second case, please indicate origin, antimicrobial tested, AMR phenotypic evaluation (MIC, KB) and why you focused on these strains.

It is not clear if WGS, MLST, ARG detection have been performed only on 23 strains. Please specify it in Materials and methods section.

Results

paragraph 2.1 MLST and phylogenetic analysis: you didn't describe the STs identified.

line 83-86: it is not a result. You can discuss the advantages of the method in the Discussion.

paragraph 2.2 Antibiotic resistance genes: I suggest to describe ARGs identified in the strain collection at first. Consideration about AST - susceptibility behaviour (line 95-99 and line 117-120) should be relocated after ARG description. 

line 117-120: Do you mean that ARGs presence was associated with phenotypic resistance? Were strains carrying blaCTX/TEM ESBL producers? What about phenotypic carbapenem resistance? I suggest to specify if there was congruence between ARG presence and phenotypic resistance for the Highets Priority Critically Important Antimicrobials (carbapenems, cephalosporins).

line 123-125: I suggest to relocate them before line 121. How do you mined sequences for intI1? Specify it in Material and methods.

line 125: which GCs?

paragraph 2.3 PCR based identification of intI1 and GCs: Why do you perform intI1 screening using both sequencing and PCR?

line 120: which observation?

line 138: Positive correlations were observed between .... AMR-GC. Do you mean between phenotypic AMR and ARG presence?

Table 2: "Integron class": do you mean Elements identified by PCR?

Figure 4: I suggest to describe these data in the text or with a table.

Author Response

The manuscript describes genomic antimicrobial resistance profiles of E. coli of marine origin (water and molluscs), focusing on ARG, intI1, and plasmid detection. Despite the importance and soundness of the argument, the manuscript is confusing and hard to understand.

Response: Thanks for judging the manuscript as scientifically sound. We have tried to better present the revised version. We hope the reviewer will find it acceptable.

In the manuscript, authors describe E. coli as "non fecal coliforms", however they underline how sewage contaminants could affect marine environment. E. coli are usually considered faecal bacteria. I suggest to remove "non fecal coliforms" or to explain what authors refer to.

Response: E. coli is a bacteria associated with fecal contamination and we have made the required correction in the revised version.

Kindly check the capital letter, cursive, commas, etc

Response: Thanks for pointing it out, we have made editorial changes.

Title

I suggest to remove "Gram Negative", because it seems redundant as all E. coli are Gram negative bacteria.

Response: We are thankful for the pertinent comment. Title has been modified as suggested.

Abstract

line 22: substitute "prevalent" with "prevalence"

Response: Abstract has been updated.

"from the coast of Kuwait": clearly define the kind of samples analysed in the study (molluscs, water ecc)

Response: A clear concise description of the type of samples used is included in the revised manuscript.

line 23: substitute "its" with "their"

Response: ‘Corrected and rewritten.

line 24: relocate "(WGS)" after "whole genome sequencing" and before "identified"

Response: Done

line 31: GCs, insert the extended name and the acronym in brackets.

Response: Done

line 32: acronym AMR, already defined in line 16.

Response: Provided at the first place mentioned.

line 32-33: I can't understand what authors mean by "significant correlation between AMR and GCs". Do you mean between phenotypic AMR and ARG presence?

Response: Yes we mean phenotypic AMR and intl1 gene as well as phenotypic AMR and GC

Introduction

line 44-45: "These" is referred to pharmaceuticals...microplastics and atmospheric dust. However, only pharmaceuticals and heavy metals could impose selective pressure and lead to ARGs spread. Please, rephrase.

Response: The statement is rephrased.

line 58: define "conventianal". Do you mean phenotypic antimicrobial susceptibility testing?

Response: Yes, we have appended the term phenotypic before antimicrobial susceptibility testing

line 59-61: Do you refer to the water environment? Please, specify it. 

Response: Yes, it is aquatic environment. Corrected.

line 61: "conventional", as above.

Response: By conventional we mean microbiological phenotypic testing. We have clarified our statements on this aspect.

line 63-63: I'm not sure you can say that molecular methods are superior to phenotypic ones. Phenotypic and molecular methods gave different and complementary information, all important for a better comprehension of the AMR phenomenon. Please, rephrase.

Response: We have discussed the advantages and dis advantages of both.

line 64: please define AST

Response: We have provided an acronym for antimicrobial susceptibility testing.

Materials and methods

paragraph 4.1.1 Marine waters: How many samples of water do you have analysed? In which period? Please indicate how many E. coli do you select from each sample (mollusc, water), for example "From each sample, at least 2 presumptive E. coli were selected...."

Response: Information is added to section 4.1 and also added the metadata in the supplementary Table S3.

line 210: What do you mean by "n=55-60 samples"? Do you refer to samples collected in summer (55) and winter (60)? Please indicate how many E. coli do you select from each sample.

Response: This information has been added. 55-60 are the number of molluskan shells and more than 200 E. coli were isolated from the collected samples.

line 218: Are these strains collected in another study or in the present one? In the first case, why do you choose to include these strains in the study? Which are their characteristics? Which antimicrobials were they resistant to? Were they isolated from water or molluscs? Please describe their features in a separate paragraph (4.1.3). In the second case, please indicate origin, antimicrobial tested, AMR phenotypic evaluation (MIC, KB) and why you focused on these strains.

It is not clear if WGS, MLST, ARG detection have been performed only on 23 strains. Please specify it in Materials and methods section.

Response: This data is taken for a study in which more than 500 E. coli were isolated both from mollusks and seawater samples. The results of the previous studies where phenotypic susceptibility were done for all the isolated strains are presented in Al Sarawi et al., 2018 [cited in the manuscript]. The genotypic data on the strains were completely lacking and therefore a representative sample set from two seasons and three sites, were chosen for whole genome sequencing [Table S3]. The AST profiles have been provided in table 1 in the present manuscript. WGS, MLST and ARG profiling were done on 23 isolates is evident from Fig 1-3.

Results

paragraph 2.1 MLST and phylogenetic analysis: you didn't describe the STs identified.

Response: Added to the revised manuscript.

line 83-86: it is not a result. You can discuss the advantages of the method in the Discussion.

Response: We prefer to keep this section as is, as it describes the strain identification part and forms the basis of the text forward.

paragraph 2.2 Antibiotic resistance genes: I suggest to describe ARGs identified in the strain collection at first. Consideration about AST - susceptibility behaviour (line 95-99 and line 117-120) should be relocated after ARG description. 

Response: Done as suggested.

line 117-120: Do you mean that ARGs presence was associated with phenotypic resistance? Were strains carrying blaCTX/TEM ESBL producers? What about phenotypic carbapenem resistance? I suggest to specify if there was congruence between ARG presence and phenotypic resistance for the Highets Priority Critically Important Antimicrobials (carbapenems, cephalosporins).

Response: In Table 1 we had shown the comparison of genotypic and phenotypic profiles. We have now replaced it with Fig 3. We have added text describing carbapenems and cephalosporin resistance and added relevant discussion in the section below.

line 123-125: I suggest to relocate them before line 121. How do you mined sequences for intI1? Specify it in Material and methods.

Response: Information added

line 125: which GCs?

Response: Relevant information added in the text

paragraph 2.3 PCR based identification of intI1 and GCs: Why do you perform intI1 screening using both sequencing and PCR?

Response: The PCR was done for all the 593 isolates. The sequencing was done only for 23 isolates and hence these were also screened for GCs & integrons.

line 120: which observation?

Response: observation of phenotypic AMR.

line 138: Positive correlations were observed between .... AMR-GC. Do you mean between phenotypic AMR and ARG presence?

Response: The correlation was performed between phenotypic AMR, intl1 and GC for 593 isolates.

Table 2: "Integron class": do you mean Elements identified by PCR?

Response: The elements identified by integron finder. More description regarding these have been added in the text.

Figure 4: I suggest to describe these data in the text or with a table

Response: The description of Figure 4 has been provided in text para 2.3.

Reviewer 3 Report

Dear Editor,

The original manuscript entitled “Antibiotic Resistance Mediated by Gram Negative Escherichia coli in Kuwait Marine Environment as Revealed by Genomic Analysis” is appropriately well written, developed and structured by Al Sarawi et al. in suitable English with a clear structure. They investigated the identification of antibiotic resistance gene elements prevalent in E. coli strains isolated from the coast of Kuwait by whole genome sequencing. They detected different genes resistant to the drug classes of beta-lactams in the samples. The methodology is novel and the research topic is very interesting and a main concern of public health these days. My main concern is the method for antibiotic resistance gene annotation in this manuscript. ABRicate tool and the ResFinder database (CGE) is an updated tool for AMR gene annotation; however,

-       I have highly recommended to detect the point mutations associated with the antibiotic resistance genes such as parC and gyrA genes in the samples.

-       You have a huge genome data-base regarding the whole genomes of E. coli strains and you can add more genomics analysis such as plasmid-finder to investigate the mobile genetic elements associated with the transferring of AMR genes in E. coli isolates to your study.

-       Methods, Results and Discussion sections must be revised according to the above comments.

After a major revising, this manuscript must be reconsidered. 

Author Response

The original manuscript entitled “Antibiotic Resistance Mediated by Gram Negative Escherichia coli in Kuwait Marine Environment as Revealed by Genomic Analysis” is appropriately well written, developed and structured by Al Sarawi et al. in suitable English with a clear structure. They investigated the identification of antibiotic resistance gene elements prevalent in E. coli strains isolated from the coast of Kuwait by whole genome sequencing. They detected different genes resistant to the drug classes of beta-lactams in the samples. The methodology is novel and the research topic is very interesting and a main concern of public health these days. My main concern is the method for antibiotic resistance gene annotation in this manuscript. ABRicate tool and the ResFinder database (CGE) is an updated tool for AMR gene annotation; however, 

-       I have highly recommended to detect the point mutations associated with the antibiotic resistance genes such as parC and gyrA genes in the samples. 

Response: We appreciate the support to the manuscript and observations. We appreciate the reviewer’s suggestion; however, we have completely exhausted all our resources for this study and we don’t have funds for this additional analyses. However, we value the suggestion and shall certainly consider the undertaking these tools for AMR gene mutations.

-       You have a huge genome data-base regarding the whole genomes of E. coli strains and you can add more genomics analysis such as plasmid-finder to investigate the mobile genetic elements associated with the transferring of AMR genes in E. coli isolates to your study. 

Response: We would like to bring to the reviewer’s attention that sequences have been aligned with plasmid finder database and about 22 mobile genetic elements have been identified in these strains that are responsible for HGT.

A more thorough analysis has been done for integrons and associated gene cassettes and the results have been presented in the modified version. We have added two figures describing additional details.

-       Methods, Results and Discussion sections must be revised according to the above comments.

 Response: The sections have been modified and updated. References to the newly used software’s have been provided.

After a major revising, this manuscript must be reconsidered. 

Response: We hope the reviewer will find the revised version acceptable.

Round 2

Reviewer 2 Report

Abstract

line 25: substitute "Fluroquinolone" with " fluroquinolone"

lines 21-22: add "genes" after "plasmids and intl1 (class 1 integron)"

line 30: add a space between "(GCs)" and "were"

line 30: "reported in 36 and 33% of E. coli, respectively." Do you refer to mollusks and sea-water samples? Please specify it.

line 31: I suggest to substitute "between AMR -intl1 (r=0.311) and AMR-GC (r=0.188)" with "between phenotypic AMR -intl1 gene (r=0.311) and phenotypic AMR-GCs (r=0.188)".

Keywords

I suggest to add a keyword referring to the marine environment.

Introduction

line 55: "Escherichia coli is an infectious animal and human bacterium". Substitute "be" with "could be"/"could become"

line 56: "The persistence of resistant E. coli has been previously demonstrated through ....". I'm not sure "persistence" is correct. Do you mean the wide AMR profile of E. coli? Please rephrase.

line 63:  "These methods are nowadays preferred over...". These methods are complementary to AST and essential to deeply understand AMR genetic basis. Please rephrase it.

Material and Methods

Paragraphs "4.1.2. Mollusk Samples", "4.1.1. Marine waters". According to Al-Sarawi et al. [28], both seawater and mollusc samples have been collected in the previous study. This should be specified in the abstract and clearly described in M&M. For example "In our previous study we analised a collection of XX E. coli isolated from seawater and mollusk...." Sampling and isolation method should be briefly reported or you can simply mention the manuscript (Al-Sarawi et al.). Instead, AMR evaluation (MIC) should be deeply described, since AMR is the focus of the manuscript.

line 276: 593 or 598 isolates?

line 276; "selected strains". You should clearly describe in the text why you have chosen these isolates. Do they have an interesting phenotypic AMR profile?

line 277: ad a space between "to" and "a"

You should add a paragraph about statistical analysis. How do you explore homogeneity of variance, data normality, and dimensions of linearity (line 173)? How do you establish the parameters were statistically significant?

Results

line 69: please write E. coli in italics

line 77: please write Enterobacteriaceae in italics

line 79: please write Enterobacter cloacae in italics

line 82: "75E0", please check the strain name as in Figure 1 is different (7SE60)

lines 79-80: "A close relationship was observed between the isolates from spatially distant locations and varied matrices (marine waters or mollusks)." What do you mean by "distant locations"? Considering Figure 1, SE19 and SC118 isolates have been collected in the same place. The same considering SC59 and 7SE60 isolates. Please rephrase.

lines 83-84: "It was noticed that the three isolates SE111, SE181 and SE158 exist separately as a group". Please specify again that these strains belonged to Enterobacter cloacae.

lines 84-87: relocate the sentences in the Discussion. Please specify that molecular tests are technically superior to microbiological methods for discriminating bacteria.

lines 110-132: Results are hard to understand, especially lines 110-116. I suggest to describe percentages of congruence between ARGs and phenotypic resistance (also with a table) for the 23 antimicrobials, and then to focus on exceptions.

line 142: add "only" before "Isolate CW141 possessed all the genetic elements (Fig 4)".

lines 142-151: I suggest to describe general features (presence of promoter, attC, etc) of the collection (also with a table) and then focus on specific strains (explaining why) 

line 148: GC1. Which cassette is GC1?

lines 167-168: "This accounted for 36% (intl1) and 33% (GCs) of the total 167 isolates derived from both the marine waters and mollusks (Fig 6a)". What do you mean? Please rephrase.

lines 170-171: "Approximately, 49% of resistant E. coli were negative for intl1 (Fig 6b)." This sentence looks in disagreement with the previous one (" 95% and 5% of intl1 positive E. coli were resistant (n=203) and susceptible (n=13) strains respectively")

lines 171-173: relocate in M&M

line 172: link between intl1-GC-resistance". Do you mean presence?

line 175 " intl1-GC-resistance (F- 24.28, P < 0.001)": the same as above

line 176: "correlations were observed between AMR-intl1 (0.310; P < 0.001) and AMR-GC". Please add "Phenotypic AMR", intI1 presence", "GC presence". Moreover, the use of GC and ARG as synonyms could be misunderstood. I suggest to choose one and use it in all the manuscript.

Figure 5: why is this figure important? Is it essential for the comprehension of the results? 

Figure 6: I suggest to change the figure with a table to be more comprehensible.

Table 1: "Table 1. Integrons in selected E. coli isolates". Why do you select these strains?

Author Response

We thank the reviewer for the second revision. We have addressed al the comments and our responses are as follows:

Abstract

line 25: substitute "Fluroquinolone" with " fluroquinolone"

Response: Done as suggested

lines 21-22: add "genes" after "plasmids and intl1 (class 1 integron)"

Response: Done as suggested

line 30: add a space between "(GCs)" and "were"

Response: Done as suggested

line 30: "reported in 36 and 33% of E. coli, respectively." Do you refer to mollusks and sea-water samples? Please specify it.

Response: 36 and 33 is the percent of intl1 and GCs. We have rephrased our statements.

line 31: I suggest to substitute "between AMR -intl1 (r=0.311) and AMR-GC (r=0.188)" with "between phenotypic AMR -intl1 gene (r=0.311) and phenotypic AMR-GCs (r=0.188)".

Response: Done as suggested

Keywords

I suggest to add a keyword referring to the marine environment.

Response: keyword added

Introduction

line 55: "Escherichia coli is an infectious animal and human bacterium". Substitute "be" with "could be"/"could become"

Response: done

line 56: "The persistence of resistant E. coli has been previously demonstrated through ....". I'm not sure "persistence" is correct. Do you mean the wide AMR profile of E. coli? Please rephrase.

Response: Done as suggested

line 63:  "These methods are nowadays preferred over...". These methods are complementary to AST and essential to deeply understand AMR genetic basis. Please rephrase it.

Response: Sentence rephrased

Material and Methods

Paragraphs "4.1.2. Mollusk Samples", "4.1.1. Marine waters". According to Al-Sarawi et al. [28], both seawater and mollusc samples have been collected in the previous study. This should be specified in the abstract and clearly described in M&M. For example "In our previous study we analised a collection of XX E. coli isolated from seawater and mollusk...." Sampling and isolation method should be briefly reported or you can simply mention the manuscript (Al-Sarawi et al.). Instead, AMR evaluation (MIC) should be deeply described, since AMR is the focus of the manuscript.

Response: we have mentioned the samples from the previous study in the abstract. We prefer to keep the brief description on the sample processing and E. coli isolation, as intl1 and GC screening by PCR was done on all these isolates. We have also added a short description on AST profiling..

line 276: 593 or 598 isolates?

Response: Thanks for pointing it out, its 598 isolates. We have corrected the number.

line 276; "selected strains". You should clearly describe in the text why you have chosen these isolates. Do they have an interesting phenotypic AMR profile?

Response: An explanation has been provided

line 277: ad a space between "to" and "a"

Response: Rephrased

You should add a paragraph about statistical analysis. How do you explore homogeneity of variance, data normality, and dimensions of linearity (line 173)? How do you establish the parameters were statistically significant?

 Response: Information added

Results

line 69: please write E. coli in italics

line 77: please write Enterobacteriaceae in italics

line 79: please write Enterobacter cloacae in italics

Response: All the genus and family names have been italicized.

line 82: "75E0", please check the strain name as in Figure 1 is different (7SE60)

Response: Thanks for bringing to our attention. Corrected

lines 79-80: "A close relationship was observed between the isolates from spatially distant locations and varied matrices (marine waters or mollusks)." What do you mean by "distant locations"? Considering Figure 1, SE19 and SC118 isolates have been collected in the same place. The same considering SC59 and 7SE60 isolates. Please rephrase.

Response: Our apologies on the oversight. We have modified our statements.

lines 83-84: "It was noticed that the three isolates SE111, SE181 and SE158 exist separately as a group". Please specify again that these strains belonged to Enterobacter cloacae.

Response: Added E. cloacae again.

lines 84-87: relocate the sentences in the Discussion. Please specify that molecular tests are technically superior to microbiological methods for discriminating bacteria.

Response: Sentences relocated in result section and evidence to molecular testing provided in the discussion.

lines 110-132: Results are hard to understand, especially lines 110-116. I suggest to describe percentages of congruence between ARGs and phenotypic resistance (also with a table) for the 23 antimicrobials, and then to focus on exceptions.

Response: We have extensively expanded our explanations on this part as per the reviewers suggestion.

line 142: add "only" before "Isolate CW141 possessed all the genetic elements (Fig 4)".

Response: done

lines 142-151: I suggest to describe general features (presence of promoter, attC, etc) of the collection (also with a table) and then focus on specific strains (explaining why) 

Response: We have provided a general explanation of these terms in the Text. Table 1 shows the presence of these elements.

line 148: GC1. Which cassette is GC1?

Response: We got this nomenclature from the integron finder database. We assume as the promoter and the recombination sites are named as Pc_1 and attc_1, hence the name GC1, most probably the open reading frame downstream to attl1.

lines 167-168: "This accounted for 36% (intl1) and 33% (GCs) of the total 167 isolates derived from both the marine waters and mollusks (Fig 6a)". What do you mean? Please rephrase.

Response: We have revisited the entire section again and rephrased the statements to make more sense.

lines 170-171: "Approximately, 49% of resistant E. coli were negative for intl1 (Fig 6b)." This sentence looks in disagreement with the previous one (" 95% and 5% of intl1 positive E. coli were resistant (n=203) and susceptible (n=13) strains respectively")

Response: We have revisited the entire section again and rephrased the statements to make more sense.

lines 171-173: relocate in M&M

Response: Done as suggested

line 172: link between intl1-GC-resistance". Do you mean presence?

Response: We mean correlation.

line 175 " intl1-GC-resistance (F- 24.28, P < 0.001)": the same as above

Response: We mean correlation.

line 176: "correlations were observed between AMR-intl1 (0.310; P < 0.001) and AMR-GC". Please add "Phenotypic AMR", intI1 presence", "GC presence". Moreover, the use of GC and ARG as synonyms could be misunderstood. I suggest to choose one and use it in all the manuscript.

 Response: Done as suggested. Done as suggested. We prefer to keep the ARGs and GC as is. The terms have been explained at its first use.

Figure 5: why is this figure important? Is it essential for the comprehension of the results? 

Response: This figure explains the results of PCR screening of intl1 and GCs.

Figure 6: I suggest to change the figure with a table to be more comprehensible.

Response: We have already provided Table1. We prefer to keep the figure to show the locations of the respective genetic elements in the strain.

Table 1: "Table 1. Integrons in selected E. coli isolates". Why do you select these strains?

Response: We have not selected these strains. This is the summarized version of the output we get from the integron finder. These elements were only found in these strains.

We hope the manuscript is now acceptable to the reviewer 

Reviewer 3 Report

Dear Editor, 

All revisions have been addressed successfully and this interesting manuscript can be considered for publishing in its present form. 

Author Response

We are thankful to the reviewer for accepting our manuscript.

Round 3

Reviewer 2 Report

Abstract

line 33: add a space between "in" and "36%".

lien 26: "within the region". Do you mean within the genome? Please rephrase or specify which region

line 34: GC, provide the extended name and the acronym in brackets

Introduction

line 63:  "less accurate". You can't say "less accurate" as phenotypic AMR assays provide different information than AMR genotypic evaluation. Genomic methods are complementary to AST and essential to deeply understand AMR genetic basis. Please rephrase it.

lines 64-66: please rephrase in accordance with the previous comment

Materials and methods

lines 278-280: also phenotypic AMR evaluation was performed in the previous study. Please specify it in this paragraph. Specify also which analysis have been performed in the present study.

line 285: Six sea-water samples. Not 8? The sites are 4 (Al-Ghazali, Al Salam, Abu Al-Hasaniya, AlKhairan) x 2 (summer and winter).

line 302: "for further processing" add bibliography [28]

lines 325-326 "These isolates were also chosen to represent varied spatio-temporal variability". What do you mean? Please specify

line 369-375: relocate in a specific paragraph.

Results

General considerations:

- I suggest to add in brackets ARGs you are referring to in all the section. I.e line 133: "ARGs against this drug class" Which ARGs? etc...

- I suggest to indicate also the precise number of isolates when you refer to a percentage. For example: 15% --> 8/250 (15%) or 15% (8/250)

line 81-83 "Molecular data analyses helped to correctly identify....  E. coli and E. cloaca strains." This is not a result. Please deepen this aspect in the Discussion.

line 133: "their". Do you mean of the strain collection?Please rephrase

line 115. add "potentially" before resistant.

I suggest to discuss beta lacatam phenotypic and genotypic resistance in one paragaph. Replace and conglomerate lines 145-159 with lines 122-125.

lines 139-141: "Besides this strain CW141 although resistant phenotypically to beta-lactams and sulfonamide did not have the respective ARGs". You already describe sulphonamide ARG abscence in the lines above. You can discuss beta-lactam ARG abscence in beta lactam paragraph (see comment above)

lines 116-121: As the result section is divided in paragraphs according to antimicrobial agents, I suggest to describe discordance between phenotypic and genotypic AMR profile in SC117, SC89 and 8KHE1 strains according to antimicrobial molecus. I.e.: discordance in beta-lactams phenotypic and genotypic profile should be discuss in beta lactam section, the same for aminoglycoside profile.

lines 153-115 "TEM-beta lactamase was one of the most common gene detected in the tested isolates" Which is the percentage?

line 167-171: it's not a result. Please relocate in Introduction or Discussion sections

lines 171-174. You should briefly summarise how many strains carry attC, intI, Pc_1  etc. Then, you can focus on specific strains (CW141, CW138 etc) explaining why you describe them. All the strains you cited in the paragraph should be reported in Table 1. It should be better to remove"selected" in table's title.

line 199: "intl1 positive isolates (n=216) and interestingly 95% of them were resistant and the remaining 5% were susceptible (Fig 6c)". Resistant/Susceptible to which antimicrobials? With "resistant" do you mean phenotypically reisistant to at least one antimicrobial? Please specify it

Author Response

Thank you for once again reviewing our manuscript. The submission has now been significantly improved. A point-to-point response to the comments has been provided in the attached document
